# ConvBERT: Improving BERT with Span-based Dynamic Convolution

**Zihang Jiang**[1][*][†], **Weihao Yu**[1][†], **Daquan Zhou**[1], **Yunpeng Chen**[2], **Jiashi Feng**[1], **Shuicheng Yan**[2]
[1]National University of Singapore, [2]Yitu Technology
jzihang@u.nus.edu, {weihaoyu6,zhoudaquan21}@gmail.com,
yunpeng.chen@yitu-inc.com, elefjia@nus.edu.sg, shuicheng.yan@yitu-inc.com

## Abstract

Pre-trained language models like BERT and its variants have recently achieved impressive performance in various natural language understanding tasks. However, BERT heavily relies on the global self-attention block and thus suffers large memory footprint and computation cost. Although all its attention heads query on the whole input sequence for generating the attention map from a global perspective, we observe some heads only need to learn local dependencies, which means the existence of computation redundancy. We therefore propose a novel span-based dynamic convolution to replace these self-attention heads to directly model local dependencies. The novel convolution heads, together with the rest self-attention heads, form a new mixed attention block that is more efficient at both global and local context learning. We equip BERT with this mixed attention design and build a ConvBERT model. Experiments have shown that ConvBERT significantly outperforms BERT and its variants in various downstream tasks, with lower training costs and fewer model parameters. Remarkably, ConvBERTBASE model achieves 86.4 GLUE score, 0.7 higher than ELECTRABASE, using less than $1/4$ training cost. Code and pre-trained models will be released [3] .

## 1  Introduction

Language model pre-training has shown great power for improving many natural language processing tasks [53, 43, 42, 25]. Most pre-training models, despite their variety, follow the BERT [10] architecture heavily relying on multi-head self-attention [52] to learn comprehensive representations. It has been found that 1) though the self-attention module in BERT is a highly non-local operator, a large proportion of attention heads indeed learn local dependencies due to the inherent property of natural language [23, 1]; 2) removing some attention heads during fine-tuning on downstream tasks does not degrade the performance [32]. The two findings indicate that heavy computation redundancy exists in the current model design. In this work, we aim to resolve this intrinsic redundancy issue and further improve BERT w.r.t. its efficiency and downstream task performance. We consider such a question: can we reduce the redundancy of attention heads by using a naturally local operation to replace some of them? We notice that convolution has been very successful in extracting local features [27, 24, 46, 16], and thus propose to use convolution layers as a more efficient complement to self-attention for addressing local dependencies in natural language.

Specifically, we propose to integrate convolution into self-attention to form a mixed attention mechanism that combines the advantages of the two operations. Self-attention uses all input tokens to generate attention weights for capturing global dependencies, while we expect to perform local

---

[*]Work done during an internship at Yitu Tech.

[†]Equal contribution.

[3]https://github.com/yitu-opensource/ConvBert

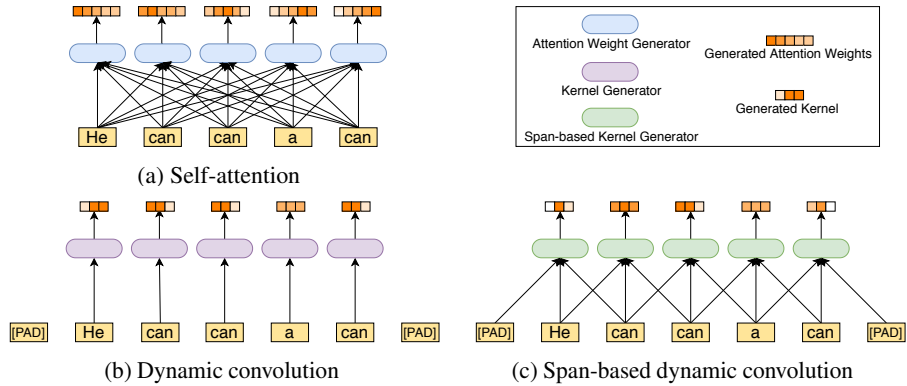

(a) Self-attention

(b) Dynamic convolution

(c) Span-based dynamic convolution

Figure 1: Processes of generating attention weights or convolution kernels. (a) Self-attention: all input tokens are needed to generate attention weights which requires quadratic complexity. (b) Dynamic convolution: dynamic kernels are generated by taking in one current token only, resulting in generating the same kernels for the same input tokens with different meanings, like "can" token. (c) Span-based dynamic convolution: kernels are generated by taking in a local span of current token, which better utilizes local dependency and discriminates different meanings of the same token (e.g., if "a" is in front of "can" in the input sentence, "can" is apparently a noun not a verb).

"self-attention", i.e., taking in a local span of the current token to generate "attention weights" of the span to capture local dependencies. To achieve this, rather than deploying standard convolution with fixed parameters shared for all input tokens, dynamic convolution [54] is a good choice that offers higher flexibility in capturing local dependencies of different tokens. As shown in Fig. 1b, dynamic convolution uses a kernel generator to produce different kernels for different input tokens [54]. However, such dynamic convolution cannot differentiate the same tokens within different context and generate the same kernels (e.g., the three "can" in Fig. 1b).

We thus develop the span-based dynamic convolution, a novel convolution that produces more adaptive convolution kernels by receiving an input span instead of only a single token, which enables discrimination of generated kernels for the same tokens within different context. For example, as shown in Fig. 1c, the proposed span-based dynamic convolution produces different kernels for different "can" tokens. With span-based dynamic convolution, we build the mixed attention to improve the conventional self-attention, which brings higher efficiency for pre-training as well as better performance for capturing global and local information.

To further enhance performance and efficiency, we also add the following new architecture design to BERT. First, a bottleneck structure is designed to reduce the number of attention heads by embedding input tokens to a lower-dimensional space for self-attention. This also relieves the redundancy that lies in attention heads and improves efficiency. Second, the feed-forward module in BERT consists of two fully connected linear layers with an activation in between, but the dimensionality of the inner-layer is set much higher (e.g., $4\times$) than that of input and output, which promises good performance but brings large parameter number and computation. Thus we devise a *grouped linear* operator for the feed-forward module, which reduces parameters without hurting representation power. Combining these novelties all together makes our proposed model, termed ConvBERT, small and efficient.

Our contributions are summarized as follows. 1) We propose a new mixed attention to replace the self-attention modules in BERT, which leverages the advantages of convolution to better capture local dependency. To the best of our knowledge, we are the first to explore convolution for enhancing BERT efficiency. 2) We introduce a novel span-based dynamic convolution operation to utilize multiple input tokens to dynamically generate the convolution kernel. 3) Based on the proposed span-based dynamic convolution and mixed attention, we build ConvBERT model. On the GLUE benchmark, ConvBERT BASE achieves 86.4 GLUE score which is 5.5 higher than BERT BASE and 0.7 higher than ELECTRA BASE while requiring less training cost and parameters. 4) ConvBERT also incorporates some new model designs including the bottleneck attention and grouped linear operator that are of independent interest for other NLP model development.

## 2 Related work

**Language model pre-traning** Language model pre-traning first pre-trains a model on large-scale unlabeled text corpora, and then fine-tunes the model on downstream tasks [8, 37, 26, 57]. It was proposed to learn separate word representations [33, 36]. Later, LSTM based CoVe [31] and ELMo [37] were developed to generate contextualized word representations. Recently, since transformer architecture including multi-head self-attention and feed-forward modules [52] has shown better effectiveness than LSTMs in many NLP tasks, GPT [40] deploys transformer as its backbone for generative pre-training and achieves large performance gain on downstream tasks. To further improve pre-trained models, more effective pre-training objectives have been developed, including Masked Language Modeling and Next Sentence Prediction from BERT [26], Generalized Autoregressive Pretraining from XLNet [57], Span Boundary Objective from SpanBERT [19], and Replaced Token Detection from ELECTRA [6]. Some other works compress the pretrained models by weight pruning [15, 12], weight sharing [26], knowledge distillation [44, 18, 50] and quantization [58, 45]. Our method is orthogonal to the above methods. There are also some works that extend pre-training by incorporating knowledge [61, 38, 28, 56], multiple languages [17, 7, 4], and multiple modalities [30, 48, 3, 49, 5]. However, to the best of our knowledge, since GPT, there is no study on improving pre-trained models w.r.t. backbone architecture design. This work is among the few works that continues the effort on designing better backbone architecture to improve pre-trained model performance and efficiency.

**Convolution in NLP models** The convolution block has been used in NLP models to encode local information and dependency of the context [59, 22, 21, 11, 60], but not explored in the pre-training field. For instance, 1D convolution is applied to specific sequence-to-sequence learning tasks [13], like machine translation and summarization. The depth-wise separable convolution is deployed in the text encoder and decoder for translation task [20], to reduce parameters and computation cost. A more recent work [54] utilizes the light-weight and dynamic convolution to further enhance the expressive power of convolution. However, all these models are limited in capability of capturing the whole context of long sentences. To enhance it, some works [47, 55] combine convolution with transformer in the sequential or multi-branch manner. To the best of our knowledge, our work is the first to explore applying convolution to pre-trained models.

## 3 Method

We first elaborate how we identify the redundancy of self-attention heads at learning local dependencies by revisiting self-attention and dynamic convolution. Then we explain the novel span-based dynamic convolution that models local dependencies and finally our proposed ConvBERT model built by the mixed attention block.

### 3.1 Motivation

**Self-attention** The self-attention block is the basic building block of BERT, which effectively models global dependencies in the input sequence of tokens. As shown in Fig. 3a, given the input $X \in \mathbb{R}^{n \times d}$ where $d$ is the hidden dimension and $n$ is the number of tokens, the self-attention module applies three linear transformations on the inputs $X$ and embeds them to the key $K$, query $Q$ and value $V$ respectively, where $K, Q, V \in \mathbb{R}^{n \times d}$. Suppose there are $H$ self-attention heads. The key, query and value embeddings are uniformly split into $d_k = d/H$-dimensional segments. The self-attention module gives outputs in the form:

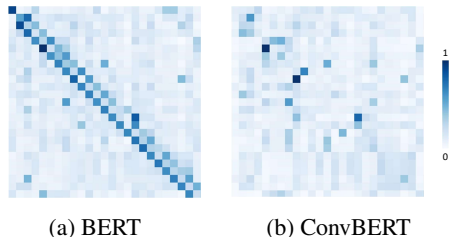

(a) BERT          (b) ConvBERT

Figure 2: Visualization of average attention map from self-attention in BERT and our ConvBERT. The example is randomly sampled from MRPC development set. The tokenized sentence is "[CLS] he said the foods ##er ##vic ##e pie business doesn ' t fit the company ' s long - term growth strategy . [SEP]".

$$\text{Self-Attn}(Q, K, V) = \text{softmax}\left(\frac{Q^\top K}{\sqrt{d_k}}\right)V. \quad (1)$$

BERT [10] and its variants successfully apply self-attention and achieve high performance. However, despite non-local essentials of the self-attention operator, some attention heads in BERT indeed learn the local dependency of the input sequence due to the inherent property of natural language [23, 1]. As shown in Fig. 2a, the averaged attention map of BERT apparently exhibits diagonal patterns similar

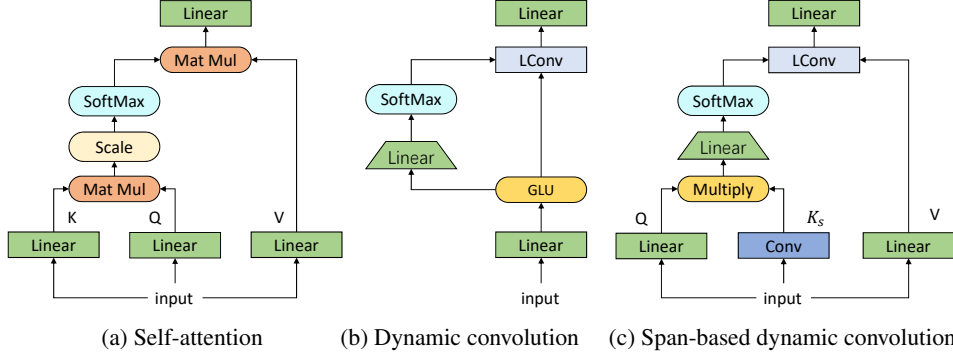

(a) Self-attention      (b) Dynamic convolution      (c) Span-based dynamic convolution

Figure 3: Illustration of self-attention, dynamic convolution [54] and proposed span-based dynamic convolution. Here LConv denotes the light-weight depth-wise convolution that ties all weights along channel dimension but uses a different convolution kernel at each position.

to the analysis of [23], meaning a large proportion of attention heads learn local dependency. Since the self-attention computes attention weights between all token pairs as in Eqn. 1, many attention weights beyond the captured local context are unnecessary to compute as they contribute much less compared to the local ones. This leads to unnecessary computation overhead and model redundancy. Motivated by this, we adopt convolution operation to capture the local dependency, considering it is more suitable for learning local dependencies than self-attention.

**Light-weight and dynamic convolution** Light-weight convolution [54] can efficiently model the local dependency. Let convolution kernel be denoted as $W \in \mathbb{R}^{d \times k}$. The output of depth-wise convolution at position $i$ and channel $c$ can be formulated as $\text{DWConv}(X, W_{c,:}, i, c) = \sum_{j=1}^{k} W_{c,j} \cdot X_{(i+j-\lceil \frac{k+1}{2} \rceil), c}$. By tying the weight along the channel dimension, we can simplify the convolution kernel to $W \in \mathbb{R}^k$, giving the following light-weight convolution:

$$\text{LConv}(X, W, i) = \sum_{j=1}^{k} W_j \cdot X_{(i+j-\lceil \frac{k+1}{2} \rceil)}. \tag{2}$$

This largely reduces the parameter size by $d$ (256 or 768 in practice) compared to the conventional depth-wise convolution. However, after training, the kernel parameters would be fixed for any input token, not favourable for capturing diversity of the input tokens. Thus, we further consider dynamic convolution [54] that generates the convolution kernel parameters conditioned on the specific input token. As illustrated in Fig. 3b, after a linear projection and a Gated Linear Unit (GLU) [9], a dynamic convolution kernel is generated from the current input token, and applied to convolve with the nearby tokens to generate new representation embedding. Compared to standard convolution kernels that are fixed after training, dynamic convolution can better utilize the input information and generate kernel conditioned on the input token. To be specific, a position dependent kernel $W = f(X_i)$ for position $i$ is used, where $f$ is a linear model with learnable weight $W_f \in \mathbb{R}^{k \times d_k}$ followed by a softmax. We denote the dynamic convolution as

$$\text{DConv}(X, W_f, i) = \text{LConv}(X, \text{softmax}(W_f X_i), i). \tag{3}$$

Compared with self-attention, which is of quadratic computation complexity w.r.t. the input sequence length, the linear complexity dynamic convolution is more efficient and more suitable for modelling local dependency and has shown effectiveness on machine translation, language modeling and abstractive summarization tasks [54]. However, we observe its convolution kernel only depends on a single token of the input, ignoring the local context. It would hurt model performance to use the same kernel for the same token in different context which may have different meanings and relations with its contextual tokens. Thus, we propose the span-based dynamic convolution as below.

## 3.2 Span-based dynamic convolution

The span-based dynamic convolution first uses a depth-wise separable convolution to gather the information of a span of tokens as shown in Fig. 3c and then dynamically generates convolution kernels. This helps the kernel capture local dependency more effectively by generating local relation of the input token conditioned on its local context instead of a single token. Besides, to make the span-based dynamic convolution compatible with self-attention, we apply linear transformation on

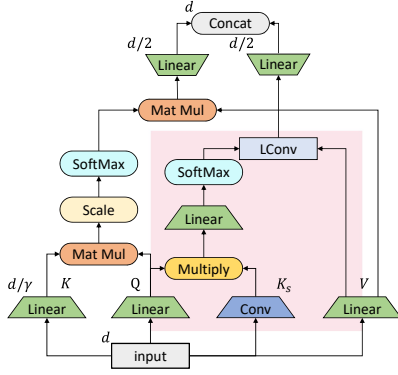

Figure 4: Illustration of mixed-attention block. It is a mixture of self-attention and span-based dynamic convolution (highlighted in pink). They share the same Query but use different Key to generate the attention map and convolution kernel respectively. We reduce the number of attention heads by directly projecting the input to a smaller embedding space to form a bottleneck structure for self-attention and span-based dynamic convolution. Dimensions of the input and output of some blocks are labeled on the left top corner to illustrate the overall framework, where $d$ is the embedding size of the input and $\gamma$ is the reduction ratio.

the input $X$ to generate query $Q$ and value $V$, and a depth-wise separable convolution to generate the span-aware $K_s$. The result of point-wise multiplication of query $Q$ and span-aware key $K_s$ pairs transformed from input $X$ is then used to generate the dynamic convolution kernel. Specifically, with query and key pair $Q, K_s$ as input, the kernel is generated by

$$f(Q, K_s) = \text{softmax}(W_f(Q \odot K_s)), \tag{4}$$

where $\odot$ denotes point-wise multiplication. As illustrated in Fig. 3c, we call this new operator span-based dynamic convolution. The output can be written as

$$\text{SDConv}(Q, K_s, V; W_f, i) = \text{LConv}(V, \text{softmax}(W_f(Q \odot K_s)), i). \tag{5}$$

Then a linear layer is applied for further process. If not otherwise stated, we always keep the same kernel size for the depth-wise separable convolution and span-based dynamic convolution.

### 3.3 ConvBERT architecture

With the above span-based dynamic convolutions, we develop the novel mixed attention block and the efficient ConvBERT model.

**Mixed attention**    The mixed attention block integrates the span-based dynamic convolution and self-attention to better model both global and local dependencies with reduced redundancy, as shown in Fig. 4. The self-attention and span-based dynamic convolution share the same query but use different keys as reference to generate attention maps and convolution kernels. Denote Cat( , ) as the concatenate operation. We formulate our mixed attention as

$$\text{Mixed-Attn}(K, Q, K_s, V; W_f) = \text{Cat}(\text{Self-Attn}(Q, K, V), \text{SDConv}(Q, K_s, V; W_f)). \tag{6}$$

The final outputs are fed to the feed-forward layer for further process.

**Bottleneck design for self-attention**    Some of the heads are redundant [32] and thus we propose to reduce the number of heads while introducing the span-based dynamic convolution module. We call this bottleneck structure, since the input embeddings are first projected to a lower-dimensional space and then pass through the self-attention module, as shown in Fig. 4. Specifically, in original BERT, the embedding feature of dimension $d$ is projected to query, key and value with the same dimension $d$ in the original transformer architecture by linear transformations. Instead, we project the embedding feature to a smaller space of dimension $d/\gamma$ for further process, where $\gamma > 1$ is the reduction ratio. Meanwhile, we reduce the number of attention heads by ratio $\gamma$. This largely saves computation cost within the self-attention and forces attention heads to yield more compact and useful attention information.

**Grouped feed-forward module**    The large number of parameters in the transformer model actually comes from the feed-forward module. To maintain the representation power while reducing parameters and computation cost, we propose to apply a grouped linear (GL) operator to the feed-forward module in a grouped manner which is defined as follows

$$M = \Pi_{i=0}^{g} \left[ f_{\frac{d}{g} \to \frac{m}{g}}^{i} \left( H_{[:, \, i-1:i \times \frac{d}{g}]} \right) \right], M' = \text{GeLU}(M), H' = \Pi_{i=0}^{g} \left[ f_{\frac{m}{g} \to \frac{d}{g}}^{i} \left( M'_{[:, \, i-1:i \times \frac{m}{g}]} \right) \right], \tag{7}$$

where $H, H' \in \mathbb{R}^{n \times d}$, $M, M' \in \mathbb{R}^{n \times m}$, $f_{d_1 \to d_2}(\cdot)$ indicates a fully connected layer that transforms dimension $d_1$ to $d_2$, $g$ is the group number and $\Pi$ means concatenation. This is in line with the

original multi-head attention mechanism, where the input features are split into multiple groups on embedding dimension and processed independently, and all processed features are concatenated again on the embedding dimension. This is more efficient than the fully connected layer and costs negligible performance drop.

By stacking the mixed attention and grouped feed-forward modules in an iterative manner, we build our ConvBERT model. As experimentally demonstrated in Sec. 4, ConvBERT is more light-weighted and efficient at capturing both global and local contexts with better performance.

# 4 Experiment

## 4.1 Implementation

We evaluate the effectiveness of our proposed architecture on the task of replaced token detection pre-training proposed by [6]. More details about the task and implementation are listed in Appendix.

We evaluate ConvBERT with a variety of model sizes. Following [6], for a *small-sized* model, the hidden dimension is 256 while the word embedding dimension is reduced to 128. Like the original transformer architecture, the intermediate layer size of the feed-forward module is 1024 (4 times of the hidden dimension) and we keep the number of layers as 12. For the number of attention heads, we keep it as 4 for the small-sized models. In addition, we also use a *medium-small-sized* model with 384 dimension embedding and 8 attention heads. By inserting span-based dynamic convolution and applying the grouped linear operation, the model can be reduced to a size comparable to a small-sized one while enjoying more representation power. For the *base-sized* model, we adopt the commonly used BERT-base configuration with 768 hidden dimension and 12 layers. For the number of heads, we use 12 for the base-sized model as our baseline. When applying the bottleneck structure (i.e. reducing the dimension of the hidden space for self-attention) we also reduce the head number by a factor $\gamma = 2$ to keep the size of each head stays the same.

During pre-trianing, the batch size is set to 128 and 256 respectively for the small-sized and base-sized model. An input sequence of length 128 is used to update the model. We show the results of these models after pre-training for 1M updates as well as pre-training longer for 4M updates. More detailed hyper-parameters for pre-training and fine-tuning are listed in the Appendix.

## 4.2 Pre-training and evaluation

**Pre-training dataset** The pre-training tasks largely rely on a large corpus of text data. The dataset WikiBooks originally used to train BERT [10] is a combination of English Wikipedia and BooksCorpus [62]. RoBERTa [29], XLNet [57] and ELECTRA [6] further propose to use larger corpora including OpenWebText [41, 14], STORIES [51], CCNEWS [34], ClueWeb [2] and Gigaword [35] to improve overall performance. However, some datasets like BooksCorpus [62] are no longer publicly available. In this paper, unless otherwise stated, we train the models on an open-sourced dataset OpenWebText [14, 41] (32G) to ease reproduction, which is of similar size with the combination of English Wikipedia and BooksCorpus that is used for BERT training. We also show the results of our model trained on the same data as BERT (i.e. WikiBooks) in the Appendix. More details of the corpus information and how to collect and filter the text data can be found in [39].

**Evaluation** We evaluate our model on the General Language Understanding Evaluation (GLUE) benchmark [53] as well as Question answering task SQuAD [43]. GLUE benchmark includes various tasks which are formatted as single sentence or sentence pair classification. See Appendix for more details of all tasks. SQuAD is a question answering dataset in which each example consists of a context, a question and an answer from the context. The target is to locate the answer with the given context and question. In SQuAD V1.1, the answers are always contained in the context, where in V2.0 some answers are not included in the context.

We measure accuracy for MNLI, QNLI, QQP, RTE, SST, Spearman correlation for STS and Matthews correlation for CoLA. The GLUE score is the average of all 8 tasks. Since there is nearly no single model submission on SQuAD leaderboard,[4] we only compare ours with other models on the development set. We report the Exact Match and F1 score on the development set of both v1.1 and v2.0. For fair comparison, unless otherwise stated, we keep the same configuration as in

ELECTRA [6]. In addition to BERT and ELECTRA, we also take knowledge distillation based methods including TinyBERT [18], MobileBERT [50] and DistillBERT [44] for comparison. All results are obtained by single task fine-tuning.

## 4.3 Ablation study

To better investigate each component, we add bottleneck structure, span-based dynamic convolution and grouped linear operation one by one to the original BERT [10] architecture. Additionally, we also increase the hidden dimension to show the performance gain by increasing the parameter size. Detailed configuration and results are shown in Table 1.

Table 1: Detailed configuration of different sized models. GLUE score is the average score on GLUE dev set. Here, BNK represents bottleneck structure; GL denotes grouped linear operator; SDConv is the proposed span-based dynamic convolution. Larger denotes increasing the hidden dimension and head dimension of the model.

| Model | Modification | Hidden dim | Head dim | Head | Group | Params | GLUE |
|---|---|---|---|---|---|---|---|
| BERTSMALL | | 256 | 64 | 4 | 1 | 14M | 75.1 |
| ELECTRASMALL | | 256 | 64 | 4 | 1 | 14M | 80.4 |
| ConvBERTSMALL | +BNK | 256 | 64 | 2 | 1 | 12M | 80.6 |
| | +BNK, +SDConv | 256 | 64 | 2 | 1 | 14M | **81.4** |
| ELECTRAMEDIUM-SMALL | | 384 | 48 | 8 | 1 | 26M | 82.0 |
| ConvBERTMEDIUM-SMALL | +BNK | 384 | 48 | 4 | 1 | 23M | 80.9 |
| | +BNK,+GL | 384 | 48 | 4 | 2 | 14M | 81.0 |
| | +BNK,+GL,+Larger | 432 | 54 | 4 | 2 | 17M | 81.1 |
| | +BNK,+GL,+SDConv | 384 | 48 | 4 | 2 | 17M | **82.1** |
| BERTBASE | | 768 | 64 | 12 | 1 | 110M | 82.2 |
| ELECTRABASE | | 768 | 64 | 12 | 1 | 110M | 85.1 |
| ConvBERTBASE | +BNK | 768 | 64 | 6 | 1 | 96M | 84.9 |
| | +BNK,+SDConv | 768 | 64 | 6 | 1 | 106M | **85.7** |

**Bottleneck structure and grouped linear operation** An interesting finding is that, introducing the bottleneck structure and grouped linear operation can reduce the number of parameters and computation cost without hurting the performance too much. It can even bring benefits in the small-sized model setting. This is possibly because the rest of the attention heads are forced to learn more compact representation that generalizes better.

**Kernel size** Another factor we investigate is the kernel size of the dynamic convolution. We show the results of applying different convolution kernel sizes on the small-sized ConvBERT model in Fig. 5. It can be observed that a larger kernel gives better results as long as the receptive field has not covered the whole input sentence. However when the kernel size is large enough and the receptive field covers all the input tokens, the benefit of using large kernel size diminishes. In later experiments, if not otherwise stated, we set the convolution kernel size as 9 for all dynamic convolution since it gives the best result.

**Ways to integrate convolution** We here test the different ways to integrate convolution into the self-attention mechanism. As shown in Table 2, directly adding a conventional depth-wise separable convolution parallel to the self-attention module will hurt the performance while inserting dynamic convolution gives little improvement over the baseline BERT architecture w.r.t. the average GLUE score. By further increasing the local dependency with span-based dynamic convolution, the performance can be improved by a notable margin.

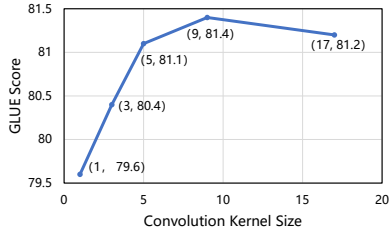

Figure 5: Results on GLUE dev set with different kernel size for span-based dynamic convolution.

## 4.4 Comparison results

We compare our ConvBERT model with BERT [10] and ELECTRA [6] as well as state-of-the-art methods [18, 44, 50, 41] on GLUE [53] and SQuAD [43] to validate the advantages of our method.

**Results on GLUE** We evaluate the performance of all methods on different downstream tasks on development set and test set of GLUE. See Table 3. Due to space limit, we only show detailed

Table 2: Comparison of ConvBERT with different convolutions. Median results of 9 runs on GLUE dev set are reported.

| Model | Convolution | MNLI | QNLI | QQP | RTE | SST-2 | MRPC | CoLA | STS-B | Avg. |
|---|---|---|---|---|---|---|---|---|---|---|
| ELECTRASMALL [6] | - | 78.9 | 87.9 | 88.3 | 68.5 | 88.3 | 87.4 | 56.8 | 86.8 | 80.4 |
| ConvBERTSMALL | Conventional | 79.9 | 85.3 | 89.2 | 63.9 | 86.9 | 83.1 | 53.4 | 83.9 | 78.2 |
| | Dynamic | 81.1 | 87.6 | 90.1 | 64.3 | 88.9 | 86.8 | 59.3 | 86.7 | 80.6 |
| | Span-based Dynamic | 81.5 | 88.5 | 90.4 | 67.1 | 90.1 | 86.8 | 59.7 | 87.7 | 81.4 |

results on test set and defer those on development set to Appendix. As can be seen from Table 3, our small and base-sized models outperform other baseline models of similar size while requiring much less pre-training cost. For example, compared with the strong baseline ELECTRABASE, our ConvBERTBASE achieves better performance with less than $1/4$ training cost. Note that TinyBERT [18] and MobileBERT [50] rely on a pre-trained large teacher network. Our model is actually a new backbone and thus is orthogonal to these compression techniques. Due to computation resource limitations, we remain the comparison of large-sized models for future work.

Table 3: Comparison of models with similar size on GLUE test set. Pre-training computation cost is also reported. [†] denotes knowledge distillation based methods relying on large pre-trained teacher models.

| Model | Train FLOPs | Params | MNLI | QNLI | QQP | RTE | SST-2 | MRPC | CoLA | STS-B | Avg. |
|---|---|---|---|---|---|---|---|---|---|---|---|
| TinyBERT[†] [18] | 6.4e19+ (49x+) | 15M | 84.6 | 90.4 | 89.1 | 70.0 | 93.1 | 82.6 | 51.1 | 83.7 | 80.6 |
| MobileBERT[†] [50] | 6.4e19+ (49x+) | 25M | 84.3 | 91.6 | 88.3 | 70.4 | 92.6 | 84.5 | 51.1 | 84.8 | 81.0 |
| ELECTRASMALL [6] | 1.4e18 (1.1x) | 14M | 79.7 | 87.7 | 88.0 | 60.8 | 89.1 | 83.7 | 54.6 | 80.3 | 78.0 |
| train longer [6] | 3.3e19 (25x) | 14M | 81.6 | 88.3 | 88.0 | 63.6 | 91.1 | 84.9 | 55.6 | 84.6 | 79.7 |
| GPT [40] | 4.0e19 (31x) | 117M | 82.1 | 88.1 | 88.5 | 56.0 | 91.3 | 75.7 | 45.4 | 66.4 | 75.9 |
| BERTBASE [10] | 6.4e19 (49x) | 110M | 84.6 | 90.5 | 89.2 | 66.4 | 93.5 | 84.8 | 52.1 | 84.8 | 80.9 |
| ELECTRABASE [6] | 6.4e19 (49x) | 110M | 85.8 | 92.7 | 89.1 | 73.1 | 93.4 | 86.7 | 59.7 | 87.7 | 83.5 |
| train longer [6] | 3.3e20 (254x) | 110M | **88.5** | 93.1 | 89.5 | 75.2 | **96.0** | 88.1 | 64.6 | **90.2** | 85.7 |
| ConvBERTSMALL | 1.3e18 (1x) | 14M | 81.5 | 88.5 | 88.0 | 62.2 | 89.2 | 83.3 | 54.8 | 83.4 | 78.9 |
| train longer for 4M updates | 5.2e18 (4x) | 14M | 82.1 | 88.5 | 88.4 | 65.1 | 91.2 | 85.1 | 56.7 | 83.8 | 80.1 |
| ConvBERTMEDIUM-SMALL | 1.5e18 (1.2x) | 17M | 82.1 | 88.7 | 88.4 | 65.3 | 89.2 | 84.6 | 56.4 | 82.9 | 79.7 |
| train longer for 4M updates | 6.0e18 (4.6x) | 17M | 82.9 | 89.2 | 88.6 | 66.0 | 92.3 | 85.7 | 59.1 | 85.3 | 81.1 |
| ConvBERTBASE | 1.4e19 (11x) | 106M | 85.3 | 92.4 | 89.6 | 74.7 | 95.0 | 88.2 | 66.0 | 88.2 | 84.9 |
| train longer for 4M updates | 5.6e19 (43x) | 106M | 88.3 | **93.2** | **90.0** | **77.9** | 95.7 | **88.3** | **67.8** | 89.7 | **86.4** |

**Results on SQuAD** We also evaluate our model on question answering task benchmark SQuAD [43]. Table 4 shows the results of our proposed ConvBERT as well as other methods [44, 18, 50, 10, 6] with similar model size. For small model size, our ConvBERTSMALL and ConvBERTMEDIUM-SMALL outperform the baseline ELECTRASMALL and achieve results comparable to BERTBASE. The results of MobileBERT are much higher since they use knowledge distillation and search for model architecture and hyper-parameters based on the development set of SQuAD. With much less pre-training cost, our base-sized model outperforms all other models of similar size.

## 5 Conclusion

We present a novel span-based dynamic convolution operator and integrate it into the self-attention mechanism to form our mixed attention block for language pre-training. We also devise a bottleneck structure applied to the self-attention module and a grouped linear operation for the feed-forward module. Experiment results show that our ConvBERT model integrating above novelties achieves consistent performance improvements while costing much less pre-training computation.

Table 4: Comparison of models with similar model size on SQuAD dev set. Pre-training computation cost is also reported. * denotes results obtained by running official code [6]. † denotes knowledge distillation based methods.

| Model | Train FLOPs | Params | SQuAD v1.1 | | SQuAD v2.0 | |
| | | | EM | F1 | EM | F1 |
|---|---|---|---|---|---|---|
| DistillBERT† [44] | 6.4e19+ (49x+) | 52M | 71.8 | 81.2 | 60.6 | 64.1 |
| TinyBERT† [18] | 6.4e19+ (49x+) | 15M | 72.7 | 82.1 | 65.3 | 68.8 |
| MobileBERT† [50] | 6.4e19+ (49x+) | 25M | 83.4 | 90.3 | 77.6 | 80.2 |
| ELECTRASMALL [6] | 1.4e18 (1.1x) | 14M | 65.6* | 73.8* | 62.8* | 65.2* |
| train longer [6] | 3.3e19 (25x) | 14M | 75.8 | 85.2* | 70.1 | 73.2* |
| BERTBASE [10] | 6.4e19 (49x) | 110M | 80.7 | 88.4 | 74.2 | 77.1 |
| ELECTRABASE [6] | 6.4e19 (49x) | 110M | 84.5 | 90.8 | 80.5 | **83.3** |
| ConvBERTSMALL | 1.3e18 (1x) | 14M | 77.1 | 84.1 | 68.7 | 70.5 |
| train longer | 5.2e18 (4x) | 14M | 78.8 | 85.5 | 71.2 | 73.6 |
| ConvBERTMEDIUM-SMALL | 1.5e18 (1.2x) | 17M | 79.4 | 86.0 | 71.7 | 74.3 |
| train longer | 6.0e18 (4.6x) | 17M | 81.5 | 88.1 | 74.3 | 76.8 |
| ConvBERTBASE | 1.4e19 (11x) | 106M | **84.7** | **90.9** | **80.6** | 83.1 |

## Broader impact

**Positive impact**   The pre-training scheme has been widely deployed in the natural language processing field. It proposes to train a large model by self-supervised learning on large corpus at first and then fine-tune the model on downstream tasks quickly. Such a pre-training scheme has produced a series of powerful language models and BERT is one of the most popular one. In this work, we developed a new pre-training based language understanding model, ConvBERT. It offers smaller model size, lower training cost and better performance, compared with the BERT model. ConvBERT has multiple positive impacts. In contrary to the trend of further increasing model complexity for better performance, ConvBERT turns to making the model more efficient and saving the training cost. It will benefit the applications where the computation resource is limited. In terms of the methodology, it looks into the model backbone designs, instead of using distillation-alike algorithms that still require training a large teacher model beforehand, to make the model more efficient. We encourage researchers to build NLP models based on ConvBERT for tasks we can expect to be particularly beneficial, such as text-based counselling.

**Negative impact**   Compared with BERT, ConvBERT is more efficient and saves the training cost, which can be used to detect and understand personal text posts on social platforms and brings privacy threat.

## Acknowledgement

We would like to thank the anonymous reviewers for their insightful comments and suggestions; Jiashi Feng was partially supported by AISG-100E-2019-035, MOE2017-T2-2-151, NUS-ECRA-FY17-P08 and CRP20-2017-0006. Weihao Yu and Zihang Jiang would like to thank TFRC program for the support of computational resources.

## Footnotes

[4] `https://rajpurkar.github.io/SQuAD-explorer/`

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
