[Supplementary Material · ConvBERT_sup_camera_ready.pdf]

# 1 Appendix

## 1.1 Datasets

### 1.1.1 GLUE dataset

GLUE benchmark introduced by [19] is a collection of nine natural language understanding tasks. The authors hide the labels of testing set and researchers need to submit their predictions to the evaluation server[1] to obtain results on testing sets. We only present results of single-task setting for fair comparison. The GLUE benchmark includes the following datasets.

**MNLI** The Multi-Genre Natural Language Inference Corpus [21] is a dataset of sentence pairs with textual entailment annotations. Given a premise sentence and a hypothesis sentence, the task is to predict their relationships including ENTENTAILMENT, CONTRADICTION and NEUTRAL. The data is from ten distinct genres of written and spoken English.

**QNLI** Question Natural Language Inference is a binary sentence pair classification task converted from The Stanford Question Answering Dataset [17], a question-answering dataset. An example of QNLI contains a context sentence and a question, and the task is to determine whether the context sentence contains the answer to the question.

**QQP** The Quora Question Pairs dataset [3] is a collection of question pairs from Quora, a community question-answering website, and the task is to determine whether a pair of questions are semantically equivalent.

**RTE** The Recognizing Textual Entailment (RTE) dataset is similar to MNLI which only has two classes, i.e., *entailment* and *not entailment*. It is from a series of annual textual entailment challenges including RTE1 [5], RTE2 [10], RTE3 [8], and RTE5 [1].

**SST-2** The Stanford Sentiment Treebank [18] is a dataset that consists of sentences from movie reviews and human annotations of their sentiment. GLUE uses the two-way (POSITIVE/NEGATIVE) class split.

**MRPC** The Microsoft Research Paraphrase Corpus [7] is a dataset from online news that consists of sentence pairs with human annotations for whether the sentences in the pair are semantically equivalent.

**CoLA** The Corpus of Linguistic Acceptability [20] is a binary single-sentence classification dataset containing the examples annotated with whether it is a grammatical English sentence.

**SST-B** The Semantic Textual Similarity Benchmark [2] is a collection of sentence pairs human-annotated with a similarity score from 1 to 5, in which models are required to predict the scores.

**WNLI** Winograd NLI [13] is a small natural language inference dataset , but as GLUE web page[2] noted, there are issues with the construction of it. Thus like previous works, GPT [15] and BERT [12] etc., we exclude this dataset for fair comparision.

### 1.1.2 SQuAD dataset

The Stanford Question Answering Dataset (**SQuAD v1.1**), a question answering (reading comprehension) dataset which consists of more than 100K questions. The answer to each question is a span of text from the corresponding context passage, meaning that every question can be answered. Then the following version **SQuAD v2.0** combines the existing data with over 50K unanswerable questions.

## 1.2 Pre-training details

We first give a brief introduction to the replaced token detection task we used for pre-training proposed by [4]. It trains the model in a discriminative way by predicting whether the token in the sequence is replaced. Meanwhile, to generate the sentence with replaced tokens as training example, they propose to use a small-sized generator trained with masked language modelling [6]. The full input sequence is first masked and then feed to the generator to get the prediction of the masked tokens. The target model then serves as a discriminator to distinguish the tokens that are wrongly predicted by the generator. The generator and the discriminator are jointly trained with masked language modelling loss and replaced token detection loss.

For the pre-training configuration, we mostly use the same hyper-parameters as ELECTRA [4]. See Table 1 for more details. While using examples of 128 sequence length for pre-training can save a lot of computation, we also find that using examples with longer sequence length can help to improve the performance on downstream task that has longer context. We pre-train our model with input sequence of length 512 for the 10% more updates before fine-tuning it for task with longer context like SQuAD. This helps the positional embedding generalize better to the downstream tasks.

Table 1: Pre-training hyper-parameters. Generator size here is the multiplier for hidden size, feed-forward inner hidden size and attention heads to compute configuration for generator. The optimizer used here is an Adam optimizer [11], and details of the optimizer are listed in the table.

| Hyper-parameter | Small | Medium-small | Base |
|---|---|---|---|
| Layer | 12 | 12 | 12 |
| Hidden dim | 256 | 384 | 768 |
| Word Embedding dim | 128 | 128 | 768 |
| feed-forward inner hidden size | 1024 | 1536 | 3072 |
| Generator size | 1/4 | 1/4 | 1/3 |
| Attention heads | 2 | 4 | 6 |
| Attention head size | 64 | 48 | 64 |
| Learning rate | 3e-4 | 5e-4 | 2e-4 |
| Learning rate decay | Linear | Linear | Linear |
| Warmup steps | 10k | 10k | 10k |
| Adam $\epsilon$ | 1e-6 | 1e-6 | 1e-6 |
| Adam $\beta_1$ | 0.9 | 0.9 | 0.9 |
| Adam $\beta_2$ | 0.999 | 0.999 | 0.999 |
| Dropout | 0.1 | 0.1 | 0.1 |
| Batch size | 128 | 128 | 256 |
| Input sequence length | 128 | 128 | 128 |

## 1.3 Fine-tuning details

Following previous work [4, 6], we search for learning rate among {5e-5, 1e-4, 2e-4, 3e-4} and weight decay among {0.01, 0.1}. For the number of training epoch, we search for the best among {10, 3}. All other parameters are kept the same as [4]. See Table 2.

## 1.4 More results

We present more results on GLUE dev set with different model sizes and pre-training settings in Table 3. As can be seen, regardless of the pre-training task and dataset size, our method consistently outperform the original BERT [6] architecture.

## 1.5 More examples and analysis of attention map

We provide more examples of the attention map in Figure 1. we also compute the diagonal concentration for the attention map $M$ as quantitative metric. It is define as $C = \frac{\sum_{|i-j| \leq 4} M_{i,j}}{\sum_{|i-j| > 4} M_{i,j}}$. This indicates

Table 2: Fine-tuning hyper-parameters. The optimizer used here is an Adam optimizer [11], and details of the optimizer are listed in the table.

| Hyper-parameter | Value |
|---|---|
| Adam $\epsilon$ | 1e-6 |
| Adam $\beta_1$ | 0.9 |
| Adam $\beta_2$ | 0.999 |
| Layer-wise LR decay | 0.8 |
| Learning rate decay | Linear |
| Warmup fraction | 0.1 |
| Dropout | 0.1 |
| Batch size | 32 |

Table 3: Comparison of our proposed ConvBERT architecture with the transformer based BERT architecture in different sizes and different pre-training settings. GLUE score represents the average score of 8 tasks on GLUE development set. MLM represents masked language modelling and RTD represents replaced token detection. The 16G WikiBooks dataset is the combination of EnWiki and BOOKCORPUS, 32G represents the OpenWebText dataset proposed by [16, 9], and 160G represents the combination of several corpus datasets used by ELECTRA [4] and RoBERTa [14]. * denotes the results from ELECTRA and $^+$ denotes the result from RoBERTa.

| Model | Pre-train task | Training data | update | Train FLOPs | Params | GLUE |
|---|---|---|---|---|---|---|
| BERTSMALL | MLM | 16G | 1.45M | 1.4e18 | 14M | 75.1* |
| | RTD | 16G | 1M | 1.4e18 | 14M | 79.7* |
| | RTD | 32G | 1M | 1.4e18 | 14M | 80.3* |
| | RTD | 160G | 4M | 3.3e19 | 14M | 81.1* |
| ConvBERTSMALL | MLM | 16G | 1.45M | 1.3e18 | 14M | 75.9 |
| | RTD | 16G | 1M | 1.3e18 | 14M | 80.6 |
| | RTD | 32G | 1M | 1.3e18 | 14M | 81.4 |
| | RTD | 32G | 4M | 5.2e18 | 14M | 81.8 |
| ConvBERTSMALL-PLUS | RTD | 32G | 1M | 1.5e18 | 17M | 82.1 |
| | RTD | 32G | 4M | 6.0e18 | 17M | 82.8 |
| BERTBASE | MLM | 16G | 1M | 6.4e19 | 110M | 82.2* |
| | MLM | 160G | 500k | 1.0e21 | 125M | 86.4$^+$ |
| | RTD | 16G | 766k | 6.4e19 | 110M | 85.1* |
| | RTD | 160G | 4M | 3.3e20 | 110M | 87.5* |
| ConvBERTBASE | RTD | 32G | 1M | 1.4e19 | 106M | 86.0 |
| | RTD | 32G | 4M | 5.6e19 | 106M | 87.7 |

how much local dependency that the attention map captures. The result in Table 4 shows that the attention in BERT concentrates more on the local dependency.

Table 4: Average concentration on MRPC.

| Model | C (diagonal-concentration) |
|---|---|
| BERT | 0.941 |
| ConvBERT | 0.608 |

## 1.6 Inference speed

We test our mixed-attention block and self-attention baseline from base-sized model on Intel CPU (i7-6900K@3.20GHz). The mixed-attention has lower Flops and is much faster than self-attention, as shown in Table 5. On the other hand, our implementation for mixed-attention on GPU and TPU is

Figure 1: More examples of attention maps.

not well optimized for the efficiency yet. Thus its acceleration may not be obvious when the input
sequence length is short. We will work on further improvement on the low-level implementation.

Table 5: Inference speed.

| Block | Flops | Speed (ms/sample) |
| --- | --- | --- |
| self-attention | 26.5G | 17.66 |
| mixed-attention | 19.3G | **12.94** |

## Footnotes

[1]https://gluebenchmark.com

[2]https://gluebenchmark.com/faq