[Reviews · NeurIPS 2020]

Review 1

Summary and Contributions: The paper introduces two ideas: 1) It lowers the parameter cost of BERT style transformer models by removing attention heads and making feed-forward modules more efficient. 2) It adds a novel 'span-based' convolution operation, that extends dynamic convolution to condition the convolution kernel on a span of tokens. Strong results on the GLUE benchmark following a pre-training scheme like ELECTRA.

Strengths: Very strong results compared to ELECTRA. Nice, if incomplete, ablation studies. An interesting approach of combining self attention and convolution, that is explained clearly. The 'span based convolution' is an interesting and novel idea.

Weaknesses: The paper introduces a complicated new module without comparing to a simple baseline (I suggest increasing the hidden dimension). There is no discussion of inference speed cost associated with the convolution operation. Update: These concerns were adressed by the authors.

Correctness: Yes.

Clarity: Yes, the paper has nice diagrams and steps the reader through the various advances in the paper.

Relation to Prior Work: Yes

Reproducibility: Yes

Additional Feedback: I think you should define, in equations, the 'Grouped feed-forward module'. Also the grouped feed-forward module and span-based convolution are basically orthogonal. The bottleneck and grouped convolution save on parameters, and you get back to 'normal' numbers of parameters by doing span-based convolution. But why can't you just increase the hidden dimension instead of introducing a complicated new module? To be fully convinced by the span-based convolution I think you need a '+BNK,+GL+larger hidden dimension' baseline in Table 1. Figure 2 shows some nice analysis suggesting the benefit of convolution, however it is just one datapoint, could you make this more rigorous? I think one of either further analysis like in figure 2, or the baseline I suggested, is needed to make this a better paper. Also I'm concerned about the added inference time of the convolution operators. I think you should include inference speed somewhere, perhaps in the ablation study, Table 1. For your broader impact statement, you should think about potential negative impacts as well as positive ones. Optional: You could test span based convolution on other NLP tasks (machine translation, etc.). I would recommend you have your submission proof-read for English style and grammar issues, if possible. Update: The author response was very strong. I have updated my score to be much higher (provided the results in the author response are added to the final paper.)


Review 2

Summary and Contributions: The paper proposes a span-based dynamic convolution to replace these self-attention heads to directly model local dependencies. The proposed heads are mixed with the original self-attention heads to reduce the FLOPs of Transformer.

Strengths: - Modifying the dynamic convolution with span-based information. - Using "grouped linear" to speedup the FFN module.

Weaknesses: - More experiments should be carried out, such as machine translation, and language modeling. The model modification should not be restricted for just BERT. - The "training cost" is only evaluated by FLOPs, while the actual training time is also very important in practice. The evaluation metrics used in [1] can be followed. - The proposed CNN-based methods have to be used together with the original self-attention module. It should be added that only using the proposed architecture. - The proposed architectures are increamental compared with previous work. It's also not novel by augmenting self-attention with CNN-based heads. [1] (ICLR) LITE TRANSFORMER WITH LONG-SHORT RANGE ATTENTION https://openreview.net/pdf?id=ByeMPlHKPH

Correctness: - The model name "ConvBERT" is confusing because the experiments are all built upon ELECTRA.

Clarity: - The paper is easy to follow and understand.

Relation to Prior Work: - The proposed architectures are increamental compared with previous work. It's also not novel by augmenting self-attention with CNN-based heads [1]. - The methods proposed in [1] should be compared with in the experiments. Their motivations and solutions are quite related. - The Figure 2 is also very similar to Figure 3 in [1]. [1] (ICLR) LITE TRANSFORMER WITH LONG-SHORT RANGE ATTENTION https://openreview.net/pdf?id=ByeMPlHKPH

Reproducibility: Yes

Additional Feedback: ====after author response I've read the author response and increased my score to 7.


Review 3

Summary and Contributions: This paper proposes ConvBERT which is based on a novel span-based dynamic convolution module aiming to learn better representations that utilize both local and global information. Compared to prior works, ConvBERT reduces the training cost by four times while achieves better performance. The method is sound to me and outperforms the previous work significantly. Fine-tuning an unsupervised pre-trained Transformer model has become the de facto approach on various NLP applications. However, the pre-training stage is notoriously slow and costly. This paper significantly reduces training time and cost and its high impact on the field is foreseeable. Therefore, I strongly recommend accept this paper. However, I believe a deeper analysis of the proposed model can be added. I am willing to increase my rating if the authors can address this concern. Update: the authors have addressed most of my concerns in the rebuttal.

Strengths: - Both the span-based dynamic convolution and the mixed attention methods are sound to me. - A 4x training time speedup compared to ELECTRA (which is already 4x faster than XLNet/RoBERTA) is very outstanding. - The authors provide a very detailed ablation study from several aspects to convince readers that performance gain is from their method.

Weaknesses: - To demonstrate that self-attention in the mixed-attention module look at global rather than local information, showing only Figure 2 is not enough. It is not clear whether this is a cherry-picked example or not. I would expect to see more examples as well as some quantitative evidences to support this claim. - Fine-tuning BERT and its variants on small datasets in GLUE are known to be unstable (Lee et al., 2020; Dodge et al., 2020). Although a recent work (Zhang et al., 2020) has shown that ELECTRA is already much more stable, it would still be more convincing to provide the mean and the standard error of multiple runs instead of showing just a single number for each setting to demonstrate that the improvement does not come from a lucky seed. - It would be better if the authors can provide the real training time of the model like the ELECTRA paper to give readers a better sense of the training cost. - Latency is one of the main concerns that hinders practitioners from using large pre-trained models such as BERT. I wonder if the proposed model has lower latency compared to the standard Transformer model during inference. References (These are not missing citations, no need to add them to the paper): [1] Lee, Cheolhyoung, Kyunghyun Cho, and Wanmo Kang. "Mixout: Effective regularization to finetune large-scale pretrained language models." ICLR (2020). [2] Dodge, Jesse, et al. "Fine-tuning pretrained language models: Weight initializations, data orders, and early stopping." arXiv preprint arXiv:2002.06305 (2020). [3] Zhang, Tianyi, et al. "Revisiting Few-sample BERT Fine-tuning." arXiv preprint arXiv:2006.05987 (2020).

Correctness: The claims and method look correct to me. Based on the statement, the authors strickly follow the experiment setup of previous works. Admittedly, I trust the authors and do not check the code and compare each hyperparameter in detail, so I cannot provide a 100% guarantee. However, there is still some space for improvement as I pointed out in the weakness section where I anticipate the authors can even do better than previous works.

Clarity: This paper is well written and clear to me. There are no obstacles when reading this paper.

Relation to Prior Work: The authors provide detailed differences compared to the previous works. It is very easy to differentiate this work from others.

Reproducibility: Yes

Additional Feedback: - Do you have any experiments supporting that mixed attention also works better when trained from scratch? Or it is only tailorred for the pre-training and then fine-tuning setting. - I wonder if the proposed mixed attention model works in the decoder setting where the convolution and self-attention are both causal. For example, replacing Transformers on language modeling and machine translation tasks.


Review 4

Summary and Contributions: The idea of this paper is motivated by the finding that a large proportion of the attention heads in BERT learn local dependencies, however self-attention is a non-local operator. A variation of the BERT model is proposed to address the redundancy issue. The main contribution is the span-based dynamic convolution operation proposed to improve the efficiency of capturing local dependency information. Instead of completely replacing self-attention with convolution, the output of self-attention and convolution are concatenated. Some minor changes, including bottleneck attention and grouped linear operator, are also used to reduce BERT parameter size.

Strengths: - First work to explore convolution for enhancing BERT efficiency. - Better performance on GLUE tasks compared with BERT and ELECTRA, while the training cost is much lower. - When the model size is small, the proposed model shows better performance on SQuAD dev sets. - Extensive ablation study to investigate impact of the changed components. Applied bottleneck attention and grouped linear operator to further reduce model size.

Weaknesses: - Although the experiments in this paper prove that span-based dynamic convolution is a sound choice to address the redundancy issue, it cannot completely replace self-attention. So the architecture of the proposed model becomes more complex. - Novelty for the span-based dynamic convolution component is not significant. - For base sized model, performance on SQuAD is not significant.

Correctness: The claims, methods and empirical methodology in the paper are generally correct. In line 139, the dimension of matrix W_f is d_k x k, which is inconsistant with equation 3. I guess the correct dimension of this matrix is k x d (similar as the one used in paper "Pay Less Attention with Lightweight and Dynamic Convolutions”)? For the span-based dynamic convolution, could you explain why you use the point-wise multiplication of K_s and Q (instead of K_s, or the concatenation of Q and K_s) in equation 4?

Clarity: The description of how the span aware key K_s is computed is not very clear (line 154), better to use equation to explain. The parameter “k” used in line 126 is different from the one used in equation 1. Better to explicitly explain what is the meaning of “k” here. Grammatical error: Line 15, “using less than 1/4 training cost”.

Relation to Prior Work: Yes, the difference is clearly discussed.

Reproducibility: Yes

Additional Feedback: You response is helpful, thanks, I expect the final version will do a good job to solve all questions and concerns of all reviewers.

[Author Response · NeurIPS 2020]

We thank all reviewers for their efforts. Below we give detailed responses.

**1. "More examples and analysis of attention map" (R1&R3)** We pro-
vide more examples of the attention map in Figure 1. we also compute
the diagonal concentration for the attention map $M$ as quantitative metric.
It is define as $C = \frac{\sum_{|i-j|\leq 4} M_{i,j}}{\sum_{|i-j|>4} M_{i,j}}$. This indicates how much local depen-
dency that the attention map captures. The result in Table 1 shows that the
attention in BERT concentrates more on the local dependency.

Figure 1: More examples of attention maps.

**2. "Inference speed" (R2&R3)** We test our mixed-attention block and self-attention baseline from base-sized model
on Intel CPU (i7-6900K@3.20GHz). The mixed-attention has lower Flops and is much faster than self-attention, as
shown in Table 2. On the other hand, for the code we submitted as the supplementary material, our implementation for
mixed-attention on GPU and TPU is not well optimized for the efficiency yet. Thus its acceleration may not be obvious
when the input sequence length is short. We will work on further improvement on the low-level implementation.

**3. "More experiments on other NLP tasks" (R1&R2&R3)** This paper focuses on improving BERT and thus the
experiments are conducted in the language pre-training scenario. Thanks for reviewers' suggestions that remind us
span based dynamic convolution/mixed attention may be applied for other NLP tasks. Due to limited time for rebuttal,
we have not finished tuning the mixed attention model (of similar size as small ConvBERT), but it has shown better
performance than transformers (of similar size as small BERT) on language modeling task on WikiText-103, as shown
in Table 3. We will explore span-based dynamic convolution/mixed attention on other tasks in the future.

| Model | C (diagonal-concentration) |
|---|---|
| BERT | 0.941 |
| ConvBERT | 0.608 |

Table 1: Average concentration on MRPC.

| Block | Flops | Speed (ms/sample) |
|---|---|---|
| self-attention | 26.5G | 17.66 |
| mixed-attention | 19.3G | **12.94** |

Table 2: Inference speed.

| Model | Perplexity |
|---|---|
| Transformer | 34.21 |
| Ours | **32.95** |

Table 3: Result on WikiText-103.

| Model | Modification | Params | GLUE |
|---|---|---|---|
| ConvBERTmedium-small | +BNK,+GL | 14M | 81.0 |
| ConvBERTmedium-small | +BNK,+GL,+Larger | 17M | 81.1 |

Table 4: Ablation study on GLUE dev set.

**4. "Actual training time" (R2&R3)** We use direct implementation of our proposed
algorithm without dedicated low-level engineering acceleration at this stage as done in
ELECTRA BERT baseline. Even such, we achieve $2.67\times$ training acceleration (from 8
days to 3 days) on the base-sized model as shown in Table 5.
22

| Model | Training time |
|---|---|
| ELECTRA-small | 12h |
| ELECTRA-base | 192h |
| CONVBERT-small | 12h |
| CONVBERT-medium-small | 18h |
| CONVBERT-base | 72h |

Table 5: Training time on TPU v3-8.

**5. "Baseline and ablation study" (R1&R2&R4)** As suggested by R1, we add a
'+BNK,+GL,+Larger' baseline that increases the hidden dimension to 432. Result are
shown in Table 4. As expected, increasing the hidden dimension only slightly improves
the result (+0.1). As suggested by R2&R4, we add the experiments of only convolution based architecture (also small
sized) which performs poorly on downstream tasks and only achieves 64 on GLUE.

**6. "Definition of Grouped feed-forward" (R1)** We will add the detailed definition in the final version. The grouped
feed-forward module is defined as follows

$$M = \Pi_{i=0}^{g}\left[f^i_{\frac{d}{g}\to\frac{m}{g}}\left(H_{[:,\ i-1:i\times\frac{d}{g}]}\right)\right], \quad M' = \text{GeLU}(M), \quad H' = \Pi_{i=0}^{g}\left[f^i_{\frac{m}{g}\to\frac{d}{g}}\left(M'_{[:,\ i-1:i\times\frac{m}{g}]}\right)\right], \quad (1)$$

where $H, H' \in \mathbb{R}^{n\times d}$, $M, M' \in \mathbb{R}^{n\times m}$, $f_{d_1\to d_2}(\cdot)$ indicates a fully connected layer that transforms dimension $d_1$ to
$d_2$, $g$ is the group number and $\Pi$ means concatenation.

**6. " Why use point-wise multiplication of $K_s$ and Q" (R4)** Instead of only using $K_s$, using point-wise multiplication
(a bi-linear operator) can merge information between a single token and its nearby tokens. Compared with concatenation
that makes the following fc layer occupy more parameters, it saves parameter number.

**7. "Comparison with Lite Transformer" (R2)** Lite transformer uses two branches of self-attention and dynamic
convolution. It is applied for translation, language modeling and abstractive summarization. While our ConvBERT
introduces span-based dynamic convolution with self-attention to form mixed-attention and is targeted for pre-training.
We have implemented 'Lite Transformer'-like architecture for pre-training. See results in row 4 'Dynamic' of Table 2
in our paper.

**8. "Mean and standard error on GLUE result" (R2)** The reported result on GLUE development set is the median of
9 runs. We list the mean and standard error of ConvBERT-small model on GLUE in Table 6.

**9. "Performance of base-sized model on**
**SQuAD" (R4)** As shown in Table 4 in our pa-
per, ConvBERT achieves similar performance
on SQuAD dataset with only $1/4$ training cost
compared to ELECTRA.

| Model | MNLI | QNLI | QQP | RTE | SST-2 | MRPC | CoLA | STS-B | Avg. |
|---|---|---|---|---|---|---|---|---|---|
| ConvBERT-small | 81.4±0.2 | 88.3±0.1 | 90.3±0.1 | 67.4±0.8 | 90.2±0.4 | 86.7±0.6 | 59.4±1.4 | 87.8±0.3 | 81.4±0.5 |

Table 6: Results on GLUE dev set.

[Meta-Review · NeurIPS 2020]

The paper proposes to replace some of the self-attention heads in Transformer/BERT models with span-based dynamic convolution. This is a good idea since many of the dependencies modeled by self-attention are local. The proposed heads are mixed with the original self-attention heads to reduce the FLOPs of Transformer. Good results are achieved on the GLUE benchmark. The authors addressed questions by the reviewers in their response and everyone agrees that this paper will be great to have at the conference.